# Biocontrol Using *Torulaspora delbrueckii* in Sequential Fermentation: New Insights into Low-Sulfite Verdicchio Wines

**DOI:** 10.3390/foods12152899

**Published:** 2023-07-30

**Authors:** Laura Canonico, Alice Agarbati, Edoardo Galli, Francesca Comitini, Maurizio Ciani

**Affiliations:** Department of Life and Environmental Sciences, Polytechnic University of Marche, Via Brecce Bianche, 60131 Ancona, Italy; l.canonico@univpm.it (L.C.); a.agarbati@univpm.it (A.A.); edogah@hotmail.it (E.G.); f.comitini@univpm.it (F.C.)

**Keywords:** wine yeasts, *Torulaspora delbrueckii*, SO_2_ reduction, bioprotection

## Abstract

*Torulaspora delbrueckii* has attracted renewed interest in recent years, for its biotechnological potential linked to its ability to enhance the flavor and aroma complexity of wine. Sequential fermentations with a selected native strain of *T. delbrueckii* (DiSVA 130) and low-sulfite native strain of *Saccharomyces cerevisiae* (DiSVA 709) were carried out to establish their contribution in biocontrol and the aroma profile. A first set of trials were conducted to evaluate the effect of the sulfur dioxide addition on pure and *T. debrueckii*/*S. cerevisiae* sequential fermentations. A second set of sequential fermentations without SO_2_ addition were conducted to evaluate the biocontrol and aromatic effectiveness of *T. delbrueckii*. Native *T. delbrueckii* showed a biocontrol action in the first two days of fermentation (wild yeasts reduced by c.a. 1 log at the second day). Finally, trials with the combination of both native and commercial *T. delbrueckii*/*S. cerevisiae* led to distinctive aromatic profiles of wines, with a significant enhancement in isoamyl acetate, phenyl ethyl acetate, supported by positive appreciations from the tasters, for ripe and tropical fruits, citrus, and balance. The whole results indicate that native *T. delbrueckii* could be a potential biocontrol tool against wild yeasts in the first phase of fermentation, contributing to improving the final wine aroma.

## 1. Introduction

In winemaking, the use of selected cultures is a suitable strategy to control the fermentation process and improve organoleptic profiles and specific aroma compounds for the production of distinctive wines [1,2]. In this regard, the use of selected non-*Saccharomyces* yeasts under suitable conditions has widened the opportunities for enhancing the specific contribution of yeasts in winemaking. Indeed, their use in mixed and sequential fermentations with the starter *Saccharomyces cerevisiae* led to an enhancement in the organoleptic qualities of wines and the complexity of aromatic notes [3,4,5]. During the last few decades, many studies have focused on the use of non-*Saccharomyces* yeasts during alcoholic fermentation for variations in several specific wine features such as an increase in glycerol [6], reduction in volatile acidity [7], enhancement in total acidity, and production of polysaccharides [8], while others focused on the enhancement in flavor and aroma complexity [9,10,11] or ethanol reduction [12]. In addition to these features, the use of non-*Saccharomyces* yeast has been proposed for biocontrol in winemaking. During the last few years there has been a trend in modern oenology to decrease sulfites because of their effect on human health. Although the World Health Organization has a recommended daily allowance (RDA) of SO_2_ of 0.7 mg SO_2/_kg of body weight, European law has set the maximum concentrations allowed at 150 mg/L and 200 mg/L in red and white wines, respectively (EU regulation no. 606/2009). Moreover, environmental concerns have led consumers to prefer “healthy” products and choose wines with lower levels of sulfites. From this perspective, the attention of winemakers was focused on research based on new strategies to reduce the use of SO_2_, which is a chemical additive with a broad spectrum and widely used in the winemaking process [13]. In this regard, in addition to chemical and physical strategies, the use of non-*Saccharomyces* yeasts could be a suitable and innovative strategy to achieve this goal along with an improvement of the aroma profile of wine. Several studies have reported the bioprotectant activity of non-*Saccharomyces* yeasts, which were found to be effective against spoilage wild microorganisms [14,15,16,17,18]. In particular, the presence of the *T. delbrueckii* strain in grape juice led to a decrease in wild yeast biodiversity if compared to the addition of sulfites [19]. 

Among the different non-*Saccharomyces* wine yeasts used in mixed fermentation with *S. cerevisiae* in winemaking, *T. delbrueckii* shows several features that positively affect the wine quality [20] and others that concern microbial interactions, such as the production of active compounds (killer toxin and hydroxytyrosol). Effectively, with respect to the attributes required to perform industrial alcoholic fermentation, among the non-*Saccharomyces* yeasts, *T. delbrueckii* is the closest species to *S. cerevisiae*. This affinity could probably be the main reason why *T. delbrueckii* was the first non-*Saccharomyces* yeast suggested for winemaking use at an industrial level.

Based on the aforementioned reasons, a selected strain of *T. delbrueckii* was used in sequential fermentation with a native *S. cerevisiae* strain already selected [21] and tested [22] for low-sulfite wine production. The aim was to evaluate the biocontrol and aroma-enhancing features of *T. delbrueckii* in organic wines using the low sulfite producer *S. cerevisiae*.

## 2. Materials and Methods

### 2.1. Yeast Strains 

The native improved strain DiSVA 709 (yeast Collection of the Department of Life and Environmental Sciences) [21] and the commercial starter strain Lalvin ICV OKAY^®^ (Lallemand Inc., Toulouse, France) were used as the *S. cerevisiae* starter strains. Both yeast strains are characterized by the absence of H_2_S production and reduced production of SO_2_. The yeast strains used in the trials were cultivated and maintained on yeast extract–peptone–dextrose (YPD) agar medium (Oxoid, Basingstoke, UK) at 4 °C for short-term storage, while for long-term storage, YPD broth supplemented with 40% (*w*/*v*) glycerol at −80 °C was used. 

### 2.2. Pilot Fermentation Trials 

The fermentation trials were carried out at Terre Cortesi Moncaro S.r.c.l. in steel vessels of 60 L containing 40 L of organic Verdicchio grape juice in duplicate under static conditions. The temperature was maintained at 18 °C. The grapes were then processed using the following procedures: soft hydraulic pressing and cold clarification at 8 °C for 2 days. The main analytical characteristics of the grape juice were pH 3.22; initial sugar content 242 g/L; total acidity 4.48 g/L; malic acid 2.3 g/L; yeast assimilable nitrogen (YAN) 60 mg/L; and total SO_2_ 14 mg/L. The YAN was adjusted to 250 mg N/L using diammonium phosphate and yeast derivative (Genesis Lift^®^ Oenofrance, Bordeaux, France). The non-*Saccharomyces* strains, *T. delbrueckii* DiSVA 130 and *T. delbrueckii* commercial strain ALPHA^®^ (Laffort, Bordeaux, Cedex), were used in the sequential fermentations after two days of the inoculum of the *S. cerevisiae* starter strains (*S. cerevisiae* DiSVA 709 and Lalvin ICV OKAY^®^) in two sets of pilot fermentation trials carried out at a winery level. The first set of trials was carried out with and without the addition 30 mg/L of SO_2_ before the inoculum of the starter strain. The other set of fermentation trials was conducted without SO_2_ added, evaluating pure and sequential fermentations using *S. cerevisiae* DiSVA 709 in sequential fermentation with *T. delbrueckii* DiSVA 130 and *T. delbrueckii* commercial strain ALPHA^®^. The fermentations were monitored by measuring the sugar consumption.

Biomass production for the inoculation of the pilot fermentation trials was carried out as follows: the yeast strain’s preculture was grown under agitation for 48 h at 25 °C (150 rpm) in modified YPD medium (0.5% yeast extract, 0.1% peptone, and 2% glucose). Five percent (*v/v*) of this preculture was inoculated in a 30 L bioreactor (Biostat® C; B. Braun Biotech Int., Goettingen, Germany) containing 25 L of the same modified YPD medium using the following conditions: 400 rpm/min; air flow of 1 vvm (L/L/min). The yeast biomass production was in feed batch modality and the biomass was collected by centrifugation and washed three times with sterile distilled water. The inoculum of the grape juice was carried out in cream form (80% humidity) at a concentration of approximately 1 × 10^6^ cell/mL. The tracking of the biomass was carried out using WL nutrient agar medium (Oxoid, Hampshire, UK) and lysine agar medium (Oxoid, Basingstoke, Hampshire, UK) [23]. The sugar consumption, measured using a Baumeé (Beé) densimeter (Polsinelli Enologia Srl, Italy), was used to monitor the fermentation process.

### 2.3. Monitoring of Yeast Population

The biomass evolution was evaluated during the fermentation using the viable cell count method. Lysine agar medium (Oxoid, Basingstoke, Hampshire, UK) was used as a selective medium that avoids the growth of *S. cerevisiae* strains, and WL nutrient agar medium (Oxoid, Basingstoke, Hampshire, UK) was used as a differential medium used for the appreciation of form, color, and diversity of wine yeast colonies. The detection of inoculated and wild yeasts in the plates was performed after incubation at 25 °C for four days. The distinction between inoculated and wild yeasts was performed using lysine agar enumeration and macro- and micro-morphological estimation of colonies in the WL nutrient agar medium. The presumptive identities of the yeasts were confirmed by sequencing using ITS 1 and 4 as the target regions. The primer pairs ITS1 (50-TCCGTAGGTGAACCTCGCG-30) and ITS4 (50-TCCTCCGCTTTATTGATATGC-30) were used to amplify the ITS1-5.8S rRNA-ITS2 region by PCR (polymerase chain reaction) following the instructions of White and co-workers [24]. The sequences obtained were compared with that provided in the GenBank database (http://www.ncbi.nlm.nih.gov/BLAST (accessed on 24 November 2022)). The inoculated *S. cerevisiae* strains and the presence of possible *S. cerevisiae* contaminant wild strains were assessed using intraspecies characterization of isolates with primer pairs δ 12/21, as described by Legras and Karst [25]. The length of the PCR products was estimated by comparing them with 100-bp marker DNA standards (GeneRuler 100-bp DNA Ladder; AB Fermentas). Ten *S. cerevisiae* isolates, confirmed by molecular methods, were then chosen for each fermentation trial.

### 2.4. Analytical Procedures 

The total acidity, volatile acidity, pH, ethanol, and total SO_2_ were analyzed following the procedures of the Official European Union Methods (EC Regulation No. 2870/00) [26]. Glucose and fructose (K-FRUGL), glycerol (K-GCROL), and malic acid (K-DMAL) were quantified using enzymatic kits (Megazyme International Ireland) according to the manufacturer instructions. A specific enzymatic kit (kit no. 112732; Roche Diagnostics, Germany) was used to determine the ammonium content. The free α-amino acids were evaluated following the protocol of Dukes and Butzke [27]. Ethyl acetate, acetaldehyde, and higher alcohols were determined using a gas chromatograph system (GC-2014; Shimadzu, Kjoto, Japan) using direct injection. In the wine samples, set up following the procedures of Canonico et al. [26], the main volatile compounds were determined using the solid-phase microextraction (HS-SPME) method [28]. The compounds were desorbed by inserting the fiber Divinylbenzene/Carboxen/Polydimethylsiloxane (DVB/CAR/PDMS) (Sigma-Aldrich, St. Louis, MO, USA) into a gas chromatograph (GC) injector. 

### 2.5. Sensory Analysis 

At the end of the fermentation, the wines, after stabilization, were bottled (750 mL) with the crown cap and maintained at 4 °C until sensory analysis. After a storage period of 3 months, the wines were subjected to sensory analysis based on smell and taste. The sensory analysis was conducted by ten tasters (80% expert and 20% non-expert), using a score scale from 1 to 9 for several descriptors (smell and taste) of each wine tested. Nine was the score of the descriptors judged to be the best while 1 was the score to be attributed in the case of very poor satisfaction. The results were used to compare the wines and provide information regarding the organoleptic quality and consumer satisfaction of the wines. The sensory analysis was carried out as follows: thirty milliliters of each wine was served at 22 ± 1 °C (room temperature) in glasses labeled with a code and covered to prevent volatile loss. The presentation order was randomized among judges.

### 2.6. Statistical Analysis 

Statistical analysis of the fermentation parameters and wine characters was conducted by analysis of variance (ANOVA) of the data of the wines. The data were analyzed using the statistical software package JMP^®^ 11. Duncan tests were used to detect the significant differences, where significance was associated with *p*-values < 0.05. 

## 3. Results

### 3.1. First Fermentation Trial: Evaluation of SO_2_ Addition in T. delbrueckii/S. cerevisiae Sequential Fermentation 

To evaluate the biocontrol effectiveness with and without the addition of SO_2_, the sequential fermentation *T. delbrueckii/S. cerevisiae* was compared with the *S. cerevisiae* pure fermentations, evaluating the biomass evolution, analytical characters, and aromatic profile. 

#### 3.1.1. Biomass Evolution and Biocontrol Activity 

In Figure 1, the results are reported of the yeast’s viable population during the inoculated fermentations with (A) and without (B) the addition of 30 mg/L of SO_2_. The *S. cerevisiae* commercial strain OKAY^®^ (Figure 1(1A)) and selected strain DiSVA 709 (Figure 1(1B)) showed that the SO_2_ addition determined a full dominance of wild yeast, while the sequential inoculation (*T. delbrueckii/S. cerevisiae*) showed a limited growth (10^5^ CFU/mL) of wild yeast on the second day and it was not detected on the ninth day. Without the addition of SO_2_, *S. cerevisiae* OKAY^®^ controls wild yeasts that slightly grow (second day) and disappear by the ninth day of fermentation. In the trial inoculated with *S. cerevisiae* DiSVA 709, wild yeasts exhibited a significant growth in the first two days compared with the *S. cerevisiae* OKAY^®^, disappearing only at the end of fermentation (Figure 1(2A)). The evolution of the wild yeasts in the sequential trial (*T. delbrueckii*/*S. cerevisiae*) (Figure 1(3B)) exhibited a similar trend to *S. cerevisiae* OKAY^®^, with a constant presence at the second day and disappearance by the ninth day. The inoculated fermentation trials with the *S. cerevisiae* starter strains (DiSVA 709 and OKAY^®^) with or without the addition of SO_2_ did not show relevant differences, with a range of occurrence of 70–90% indicating an overall dominance.

#### 3.1.2. Main Analytical Characteristics 

In Table 1 are reported the main oenological characters of the pure and sequential fermentation trials with and without SO_2_ addition. As expected, the addition of 30 mg/L of SO_2_ did not exert an influence on the main analytical characteristics of the wines. Indeed, there were no relevant differences in the main analytical compounds with or without the addition of SO_2_. Among the trials with different inoculated starter strains, there was a reduction in the final ethanol amount in the sequential fermentation *T. delbrueckii* DiSVA 130/*S. cerevisiae* DISVA 709 strain, particularly in comparison with that of the OKAY^®^ commercial strain without sugar residue (Figure 1).

#### 3.1.3. Main Volatile Compounds 

The results of the influence of sequential fermentation using *T. delbrueckii/S. cerevisiae* DiSVA 709 in comparison with pure *S. cerevisiae* DiSVA 709 and commercial strain OKAY^®^ fermentations on the volatile profiles of the wines are shown in Table 2. 

The addition of SO_2_ did not show any relevant influence on the production of volatile compounds, except for ethyl acetate. Indeed, in both pure fermentations of *S. cerevisiae* (OKAY^®^ and DiSVA 709), the absence of SO_2_ induced a production of ethyl acetate double or more if compared with the trial with SO_2_ added, while the presence of *T. delbrueckii* significantly reduced this increase. Moreover, the sequential fermentation of *T. delbrueckii/S. cerevisiae* DiSVA 709 showed a significant enhancement in ethyl butyrate, β-phenyl ethanol, and geraniol in the trials without SO_2_ added, while in the same condition, a significant increase in ethyl acetate was found in pure cultures of *S. cerevisiae* (DiSVA 709 and OKAY^®^). The sequential fermentation trials showed the highest amounts of higher alcohols (except for n-propanol) and isomyl acetate, while there were no substantial differences in terpene production. The oenological features of the OKAY^®^ strain were confirmed by Agarbati and colleagues [21], demonstrating the highest production of ethyl butyrate and n-propanol and lower production of phenylethyl acetate.

### 3.2. Second Set of Fermentation Trials: Evaluation and Comparison of Native T. delbrueckii DiSVA 130 and Commercial Strain ALPHA^®^ in Sequential Fermentation with S. cerevisiae DiSVA 709 (No SO_2_ Added)

The results from the first set of fermentations established that sequential fermentations ensured effective wild yeast control in the absence of sulfites. In these trials, with no SO_2_ added, the native strain *T. delbrueckii* DiSVA 130 was compared with a commercial strain, *T. delbrueckii* ALPHA^®^, in combination with native *S. cerevisiae* DiSVA 709.

#### 3.2.1. Biomass Evolution and Biocontrol Action 

In Figure 2, the growth kinetics of the fermentation trials are shown in a comparative assessment. The most relevant differences were found on the second day, where the combination *T. delbrueckii* DiSVA 130/*S. cerevisiae* DiSVA 709 showed a slight but significant reduction in the initial wild yeast population of about 10^4^ CFU/mL, while the other two fermentation trials showed an increase of a one-log order (Figure 2D). However, by the ninth day the wild yeasts had completely disappeared in all trials. In the second fermentation set, the *S. cerevisiae*-inoculated strain showed a dominance toward wild *S. cerevisiae* strains with 70%, 75%, and 80% of occurrence for *T. delbrueckii* ALPHA^®^/*S. cerevisiae* DiSVA 709, *T. delbrueckii* DiSVA 130/*S. cerevisiae* DiSVA 709, and *S. cerevisiae* DiSVA 709 pure fermentation, respectively.

#### 3.2.2. Main Oenological Characteristics 

As shown in Table 3, the sequential fermentation trials did not differ among them, while pure fermentation of *S. cerevisiae* DiSVA 709 resulted in a significantly higher ethanol production without sugar residue (Figure 2).

#### 3.2.3. Main Oenological Volatile Compounds 

Regarding the main volatile compounds (Table 4), significant differences were shown. Both sequential fermentations showed significantly higher amounts of esters and higher alcohols (with the exclusion of isoamylic alcohol) and terpenes in comparison with pure fermentation. The sequential fermentation of *T. delbrueckii* DiSVA 130/*S. cerevisiae* DiSVA 709 was characterized by ethyl acetate, isoamyl acetate, isobutanol amyl alcohol, linalool, and nerol production, while *T. delbrueckii* ALPHA^®^/*S. cerevisiae* DiSVA 709 exhibited significantly higher amounts of ethyl butyrate, ethyl hexanoate, isoamyl acetate, β-phenyl ethanol, phenyl ethyl acetate, linalool, and geraniol. However, the level of terpenes was lower than the threshold values in wines for these compounds. 

### 3.3. Sensory Analysis of Verdicchio Wines Inoculated with S. cerevisiae DiSVA 709, T. delbrueckii DiSVA 130/S. cerevisiae DiSVA 709, and T. delbrueckii ALPHA^®^/S. cerevisiae DiSVA 709

The wines obtained by the second fermentation trials (without SO_2_ added) were evaluated by sensory analysis to establish the role and the influence of *T. delbrueckii* in their aroma features and complexity. The results, reported in Figure 3, show a general appreciation by the tasters, with each wine distinguished by distinctive aromatic notes and without defects. The wines obtained by *T. delbrueckii/S. cerevisiae* DiSVA709 fermentation were perceived to be more balanced with relevant fruitiness (ripe fruit, tropical fruit, and citrus). The wines produced by *T. delbrueckii*/*S. cerevisiae* ALPHA^®^ and *S. cerevisiae* DiSVA 709 fermentations were perceived with the same trend but with a lower score. These results fit well with the determination of some volatile compounds as acetate esters.

## 4. Discussion

The renewed interest in non-*Saccharomyces* yeasts has led to the industrial production of selected cultures for winemaking. Currently, *T. delbrueckii* is the first non-*Saccharomyces* species produced for this purpose, and the most commercially available active dry yeast. 

The ability of *S. cerevisiae* to compete with other non-*Saccharomyces* yeasts and to dominate wine fermentation is well established. Moreover, *T. delbrueckii* generally has less fermentation vigor and a lower growth rate than *S. cerevisiae* under usual wine fermentation conditions [10,12], and this behavior may suggest a difficulty in dominating must fermentation in the presence of *S. cerevisiae* yeasts [31]. However, under the conditions tested (two days of sequential inoculation), the *T. delbrueckii* DiSVA 130 strain did not seem to be affected by the presence of *S. cerevisiae* DiSVA 709.

The results of the fermentation kinetics agreed with previous studies [7,32], reporting a lower ethanol production in the trials where the musts were inoculated with *T. delbrueckii* yeast, although no statistical differences were seen between commercial *T. delbrueckii* and the selected native strain.

Regarding the biocontrol action, Simonin et al. [19] reported noticeable bioprotectant and antioxidant effects of *T. delbrueckii* inoculated at the beginning of the white winemaking process, while Chacon-Rodriguez, et al. [15] showed a biocontrol action of a blend of *T. delbrueckii* and *Metschnikowia pulcherrima* applied to a machine harvester as compared to the standard addition of SO_2_ in the Cabernet Sauvignon variety. In agreement with Simonin et al. [19], the addition of *T. delbrueckii* DiSVA 130 showed a controlling effect over wild yeasts during the first two days of fermentation, although slightly lower if compared with the sulfites fermentation control trial. However, *T. delbrueckii* DiSVA 130 effectively limited the development of wild yeasts, demonstrating its effectiveness to protect must. Several modalities of actions that can explain the biocontrol action of some strains of *T. delbrueckii* are still to be evaluated. Some strains of *T. delbrueckii* were identified to possess the killer character [31,33,34]. On the other hand, other antimicrobial actions, such as competition for nutrients or the production of antimicrobial peptides, could be involved.

The impact of *T. delbrueckii* on fermentation and aroma enhancement has been documented over the years [35,36]. A lot of studies showed the positive contribution of *T. delbrueckii* strains and their relative positive impact on wine quality [9,20,37]. This non-*Saccharomyces* yeast is recommended for the fermentation of both dry and high-sugar grapes for the low production of acetic acid. Azzolini and coworkers [38] already demonstrated that multi-starter fermentation with *T. delbrueckii* greatly affected the content of several important volatile compounds, including ß-phenyl ethanol, isoamyl acetate, fatty acid esters, C4–C10 fatty acids, and vinyl phenols. Ramirez and Velazquez [31] analyzed the variable behavior of *T. delbrueckii* considering the strain’s differences and wine varieties, with a special emphasis on the proposals for industrial use of this species.

The production of esters by *T. delbrueckii* might be strain-dependent and it is further modified in the presence of *S. cerevisiae* during multiple fermentations [9,39]. This could explain some of the results obtained in this work concerning the volatile compounds phenyl ethyl acetate and ß-phenyl ethyl ethanol, which typically increase in the presence of *T. delbrueckii.* In the first set of trials, conducted with and without sulfur dioxide, *T. delbrueckii* DiSVA 130/*S. cerevisiae* DiSVA 709 trials showed a slight increase in phenyl ethyl acetate only, while isoamyl acetate and phenyl ethyl ethanol increased significantly only without sulfur dioxide. In the second set of the trials, both *T. delbrueckii* DiSVA 130 and the commercial strain ALPHA^®^ in sequential fermentation with *S. cerevisiae* DiSVA 709 determined a 10-fold increase in phenyl ethyl acetate compared with pure *S. cerevisiae* fermentation. On the other hand, in the conditions tested, β-phenyl ethyl ethanol increased only slightly in the *T. delbrueckii* ALPHA^®^/*S.cerevisiae* DiSVA 709 fermentation, while the presence of *T. delbrueckii* DiSVA 130 did not cause any increase. On the other hand, in agreement with Sun et al. [29], both sequential fermentations using *T. delbrueckii* DiSVA 130 and ALPHA^®^ revealed a significant enhancement in ethyl acetate and phenyl ethyl acetate contents, while the amounts of terpenes were in general lower than the threshold values. 

The overall analytical profiles of the wines did not show the presence of any defects in the presence of *T. delbrueckii*, showing, on the contrary, some differences in esters and higher alcohols, and the sensory evaluation highlighted the effective positive contribution of these non-*Saccharomyces* yeasts, particularly the native strain *T. delbrueckii* DiSVA 130, which imparts notes of tropical fruit, citrus, and ripe fruit and gives a greater balance to the wine.

The overall results indicated the multiple roles of *T. delbrueckii* in winemaking, since the selected DiSVA 130 strain showed an effective biocontrol action in sequential fermentation of Verdicchio wine in the absence of SO_2_ addition. At the same time, this fermentation modality gave a distinctive and aromatic imprint to the wine, as corroborated by the sensory analysis. 

There is a negative perception developed by consumers towards sulfites in wine, because of health and environmental concerns, that determined a new trend in the winemaking market. For this, there is increasing demand for wines with health benefits, and with low SO_2_ content, that push winemakers toward strains with tailored characteristics. 

In this research, the application of *T. delbrueckii* DiSVA 130 in sequential fermentation with native *S. cerevisiae* DiSVA 709 demonstrated a biocontrol activity in the absence of SO_2_, revealing a synergistic effect of two native strains to impart distinctive aromatic notes to wines.

## Figures and Tables

**Figure 1 foods-12-02899-f001:**
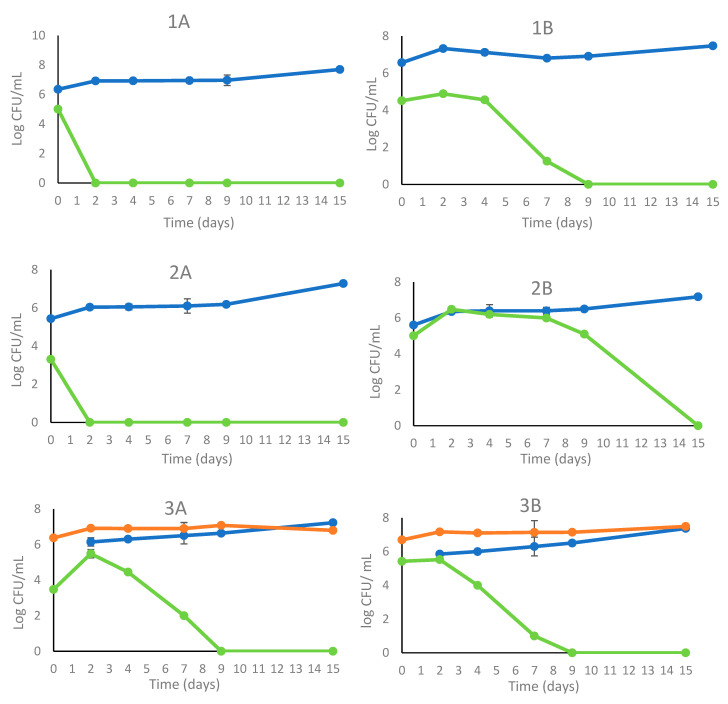
Growth kinetics in sequential fermentation trials of *S. cerevisiae* commercial strain OKAY® (1) (
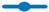
), *S. cerevisiae* DiSVA 709 (2) (
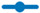
), *T. delbrueckii* DiSVA 130 (

)/*S*. *cerevisiae,* DiSVA 709 (
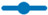
), and wild yeast (
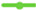
) (3) on natural grape juice with (**A**) and without (**B**) SO_2_.

**Figure 2 foods-12-02899-f002:**
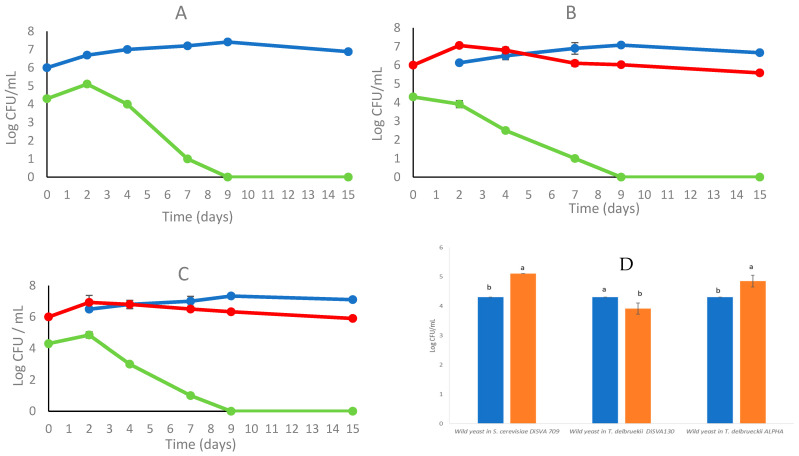
Growth kinetics and biocontrol action of *S. cerevisiae* DiSVA 709 in pure culture (**A**) in sequential fermentation with *T. delbrueckii* strain DiSVA 130 (**B**) and commercial strain *T. delbrueckii* ALPHA^®^ (**C**) (without SO_2_ addition). *S. cerevisiae* DiSVA 709 (
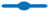
), wild yeasts (
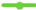
), *T. delbrueckii* (
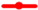
) on Verdicchio grape juice. (**D**) Effect of different fermentations on wild yeast population at inoculation time 0 (
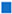
) and after two days (
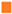
). Data with different superscript letters (a,b) are significantly different (Duncan tests; *p* < 0.05).

**Figure 3 foods-12-02899-f003:**
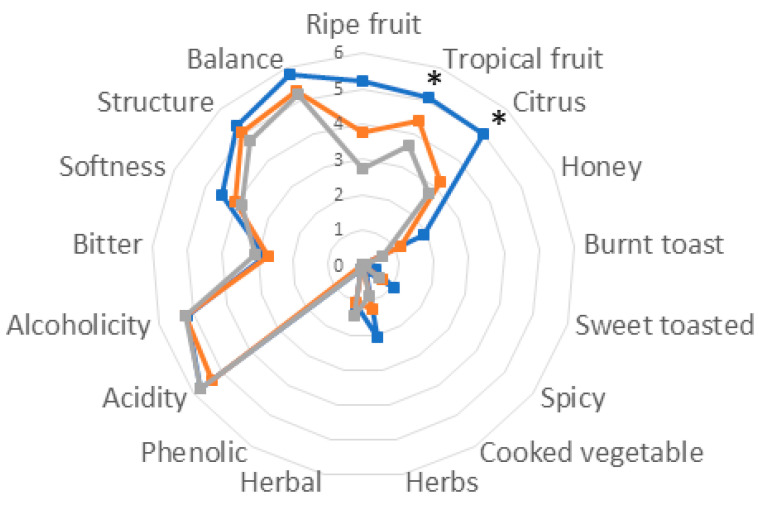
Sensory analysis of Verdicchio wines inoculated with *S. cerevisiae* DiSVA 709 (
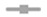
); *T. delbrueckii* DiSVA 130/*S. cerevisiae* DiSVA 709 (
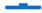
); *T. delbrueckii* ALPHA^®^/*S. cerevisiae* DiSVA 709 (
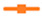
). * Significant difference at *p* = 0.05.

**Table 1 foods-12-02899-t001:** Main oenological characters of *T. delbrueckii* in sequential fermentation with and without SO_2_ added as compared to fermentation with pure *S. cerevisiae* starter strains. Data are the means ± standard deviation. Data with different superscript letters (a,b,c) within each column are significantly different (Duncan tests; *p* < 0.05).

Inoculated Strains	Ethanol (%*v/v*)	Total Acidity (g/L)	Volatile Acidity (g/L)	Malic Acid (g/L)
*S. cerevisiae* OKAY^®^ + SO_2_	14.69 ± 0.01 ^a^	5.68 ± 0.02 ^c^	0.30 ± 0.00 ^a^	0.9 ± 0.00 ^b^
*S. cerevisiae* OKAY^®^	14.88 ± 0.02 ^a^	5.27 ± 0.01 ^c^	0.31 ± 0.01 ^a^	0.7 ± 0.00 ^b^
*S. cerevisiae* DiSVA 709 + SO_2_	14.05 ± 0.12 ^b^	6.29 ± 0.05 ^a^	0.22 ± 0.01 ^a^	1.25 ± 0.07 ^a^
*S. cerevisiae* DiSVA 709	14.35 ± 0.1 ^b^	5.77 ± 0.00 ^ab^	0.30 ± 0.01 ^a^	1.00 ± 0.00 ^ab^
*T. delbrueckii* DiSVA 130/*S. cerevisiae* DiSVA 709 + SO_2_	13.9 ± 0.02 ^c^	6.14 ± 0.14 ^a^	0.25 ± 0.00 ^a^	1.35 ± 0.07 ^a^
*T. delbrueckii* DiSVA 130/*S. cerevisiae* DiSVA 709	13.86 ± 0.09 ^c^	5.35 ± 0.03 ^c^	0.29 ± 0.00 ^a^	1.45 ± 0.07 ^a^

**Table 2 foods-12-02899-t002:** The main volatile compounds of *T. delbrueckii* in sequential fermentation as compared with pure fermentation of *S. cerevisiae* starter strain (mg/L). The threshold values are reported in brackets (mg/L). Data are the means ± standard deviation. Data with different superscript letters (a,b,c) within each column are significantly different (Duncan tests; *p* < 0.05).

	OKAY^®^ + SO_2_	OKAY^®^	*S. cerevisiae* DiSVA 709 + SO_2_	*S. cerevisiae* DiSVA 709	*T. delbrueckii* DiSVA 130/*S. cerevisiae* DiSVA 709 + SO_2_	*T. delbrueckii* DiSVA 130/*S. cerevisiae* DiSVA 709
**ESTERS**						
Ethyl butyrate (0.02)	1.214 ± 0.021 ^b^	1.491 ± 0.075 ^a^	0.653 ± 0.223 ^c^	0.628 ± 0.035 ^c^	0.208 ± 0.011 ^d^	0.441 ± 0.084 ^cd^
Ethyl acetate (7.50)	19.71 ± 2.17 ^b^	36.35 ± 1.61 ^a^	10.439 ± 0.68 ^b^	38.201 ± 0.86 ^a^	14.18 ± 1.06 ^b^	19.75 ± 0.70 ^b^
Ethyl hexanoate (0.014)	1.063 ± 0.2354 ^a^	0.16 ± 0.0017 ^b^	0.253 ± 0.0974 ^b^	0.191 ± 0.0110 ^b^	0.081 ± 0.0149 ^b^	0.161 ± 0.0464 ^b^
Isoamyl acetate (0.03)	0.947 ± 0.042 ^b^	0.852 ± 0.034 ^b^	1.357 ± 0.462 ^b^	1.095 ± 0.026 ^b^	1.425 ± 0.134 ^b^	3.331 ± 0.375 ^a^
Phenyl ethyl acetate (0.25)	0.31 ± 0.01 ^b^	0.27 ± 0.04 ^b^	0.64 ± 0.01 ^a^	0.76 ± 0.04 ^a^	0.63 ± 0.13 ^a^	0.79 ± 0.19 ^a^
**ALCOHOLS**						
n-propanol (9.0)	86.630 ± 0.94 ^a^	94.148 ± 1.51 ^a^	39.655± 0.26 ^b^	37.032 ± 2.64 ^b^	27.904 ± 0.71 ^b^	35.734 ± 2.10 ^b^
Isobutanol (40.0)	17.51± 0.18 ^b^	12.56 ± 0.63 ^b^	10.957± 2.02 ^b^	19.211 ± 0.52 ^b^	32.634 ± 0.04 ^a^	25.211± 0.85 ^a^
Amyl alcohol (2.2)	12.601 ± 2.27 ^b^	12.245 ± 1.51 ^c^	19.211 ± 0.51 ^c^	14.909± 0.08 ^b^	20.74 ± 1.50 ^a^	25.690 ± 0.92 ^a^
Isoamyl alcohol (30.0)	132.53 ± 2.18 ^b^	145.105 ± 1.57 ^b^	137.156 ± 0.99 ^b^	125.50± 0.13 ^b^	171.56± 2.71 ^a^	192.248 ± 1.68 ^a^
β-Phenyl ethanol (14.0)	13.93 ± 0.09 ^bcd^	10.05 ± 0.17 ^cd^	18.90 ± 0.03 ^ab^	15.83 ± 0.21 ^bc^	8.02 ± 0.20 ^d^	25.42 ± 0.65 ^a^
**CARBONYL COMPOUNDS**						
Acetaldehyde (0.5)	6.40 ± 2.83 ^ab^	3.23 ± 0.15 ^b^	8.22 ± 0.31 ^a^	7.70 ± 1.85 ^a^	4.79 ± 0.50 ^ab^	3.02 ± 0.68 ^b^
**TERPENES**						
Linalool (0.025)	0.197 ± 0.079 ^a^	0.125 ± 0.064 ^a^	0.153 ± 0.1209 ^a^	0.186 ± 0.132 ^a^	0.078 ± 0.014 ^a^	0.128 ± 0.012 ^a^
Geraniol (0.030)	0.009 ± 0.0005 ^abc^	0.007 ± 0.003 ^bc^	0.016 ± 0.003 ^a^	0.013 ± 0.004 ^ab^	0.003 ± 0.003 ^c^	0.013 ± 0.005 ^ab^
Nerol (0.025)	0.006 ± 0.003 ^ab^	0.004 ± 0.006 ^ab^	ND **	0.009 ± 0.00^2 ab.^	0.011 ± 0.004 ^a^	0.008 ± 0.002 ^ab^

Threshold values from [29,30]; ** = Not detected.

**Table 3 foods-12-02899-t003:** The main analytical characteristics of pure fermentation of *S. cerevisiae* DiSVA 709 and sequential fermentation with *T. delbrueckii* DiSVA 130 and ALPHA^®^ commercial strain. Data with different superscript letters (a,b) within each column are significantly different (Duncan tests; *p* < 0.05).

	Ethanol (%*v/v*)	Total Acidity (g/L)	Volatile Acidity (g/L)	Malic Acid (g/L)
*S. cerevisiae* DiSVA 709	14.43 ± 0.00 ^a^	5.52 ± 0.02 ^a^	0.25 ± 0.06 ^a^	1.2 ± 0.00 ^a^
*T. delbrueckii* DiSVA 130/*S. cerevisiae* DiSVA 709	13.77 ± 0.10 ^b^	5.55 ± 0.14 ^a^	0.23 ± 0.01 ^a^	1.2 ± 0.14 ^a^
*T. delbrueckii* ALPHA^®^/*S. cerevisiae* DiSVA 709	13.71 ± 0.02 ^b^	5.52 ± 0.06 ^a^	0.25 ± 0.03 ^a^	1.1 ± 0.00 ^a^

**Table 4 foods-12-02899-t004:** The main volatile compounds of pure fermentation of *S. cerevisiae* DiSVA 709 and sequential fermentation with *T. delbrueckii* DiSVA 130 and ALPHA^®^ commercial strain (mg/L). The threshold values are reported in brackets (mg/L). OAV: odor activity value. ND = not detected. Data with different superscript letters (a,b,c) within each column are significantly different (Duncan tests; *p* < 0.05).

	*S. cerevisiae* DiSVA 709 (mg/L)	OAV	*T. delbrueckii* DiSVA 130/*S. cerevisiae* DiSVA 709 (mg/L)	OAV	*T. delbrueckii* ALPHA^®^/*S. cerevisiae* DiSVA 709 (mg/L)	OAV
**ESTERS**						
Ethyl butyrate (0.02)	0.40 ± 0.10 ^b^	1	0.31 ± 0.01 ^c^	0.77	0.52 ± 0.19 ^a^	1.3
Ethyl acetate (7.50)	26.42 ± 4.29 ^b^	2.2	59.88 ± 2.14 ^a^	4.99	33.67 ± 6.71 ^b^	2.8
Ethyl hexanoate (0.014)	2.76 ± 0.33 ^b^	34.5	2.90 ± 0.41 ^ab^	36.25	3.39 ± 0.35 ^a^	42.37
Isoamyl acetate (0.03)	0.90 ± 0.01 ^b^	5.62	0.95 ± 0.08 ^a^	4.06	1.03 ± 0.04 ^a^	6.43
Phenylethyl acetate (0.25)	0.08 ± 0.01 ^c^	1.09	0.74 ± 0.16 ^b^	10.13	0.98 ± 0.08 ^a^	13.42
**ALCOHOLS**						
n-propanol (9.0)	37.01 ± 3.09 ^a^	0.12	39.25 ± 1.40 ^a^	0.13	38.73 ± 0.79 ^a^	0.12
Isobutanol (40.0)	15.38 ± 1.72 ^b^	0.38	26.67 ± 5.20 ^a^	0.66	11.92 ± 3.30 ^c^	0.30
Amyl alcohol (12.2)	12.99 ± 0.26 ^b^	0.20	39.76 ± 8.28 ^a^	0.62	14.33 ± 3.77 ^b^	0.22
Isoamyl alcohol (30.0)	123.34 ± 8.0 ^a^	2.05	67.11 ± 5.45 ^b^	1.11	126.30 ± 2.7 ^a^	0.47
β-Phenyl ethanol (14.0)	7.45 ± 0.01 ^b^	0.53	7.40 ± 0.16 ^b^	0.52	9.10 ± 0.02 ^a^	0.65
**CARBONYL COMPOUNDS**						
Acetaldehyde (0.50)	19.23 ± 0.50 ^a^	38.46	14.25 ± 0.27 ^b^	28.5	14.23 ± 2.87 ^b^	28.46
**MONOTERPENES**						
Linalool (0.025)	0.03 ± 000 ^b^	1.2	0.20 ± 0.07 ^a^	8	0.22 ± 0.14 ^a^	8.8
Geraniol (0.030)	ND	0	0.006 ± 0.00 ^a^	0.2	0.003 ± 0.00 ^b^	0.1
Nerol (0.025)	0.003 ± 0.001 ^b^	0.2	0.004 ± 0.00 ^b^	0.26	0.006 ± 0.00 ^a^	0.4

## Data Availability

The data used to support the findings of this study can be made available on request from the corresponding author. Data is contained within the article.

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
