# Peer review of "Biocontrol Using Torulaspora delbrueckii in Sequential Fermentation: New Insights into Low-Sulfite Verdicchio Wines"

_foods, 2023, doi:10.3390/foods12152899_

Round 1
Reviewer 1 Report
Line 64: a bit pedantic but I would not consider T. delbrueckii to be the “most closed” species to S. cerevisiae as there are now also a strong focus on Kazachstania spp. in some wine labs. Please reword.
Line 80: is 80% really the final concentration. It is unusually high? Please clarify.
Line 85: “then” not “the”
Line 111: Please add more details of the manufacturer of the Baumé (◦Bé) densimeter
Line 114: shouldn’t it be non-S. Cerevisiae strains? Normally S. cerevisiae cannot grow on this medium
Line 120: how many colonies were screened with PCR?
Line 179: duplication of “in the”
Table 1: the 5.685±0.02c, 0.31±0.014a, 5.355±0.03c, 0.225±0.007a and 0.305±0.007a values have three decimal points and the 5.27±0,01c value has a comma
Table 2: in “brackets”, hexanoate, isoamyl alcohol, terpenes (similar typos also in Table 4)
Line 221: terpenes throughout the manuscript is wrongly spelled
Section 3.2. Species names not in cursive
Table 3: Values 0.25±0.028. 0.23±0.007 and 0.25±0.063 have three decimal points
Figure 3: Please rearrange the order of the descriptors within the spider chart (sensory analysis) as is is a bit difficult to interpret, for instance between “phenolic and sweet toasted it is difficult to know which is which.
Line 284: please rephrase the line, it sounds clumsy
Line 291: “affected”
some references are missing DOIs (like 9 and 12, 23), 15, 27 have some word starting with a capital letter
English is fine.
minor errors that would be corrected by the text editor
Author Response
Reviewer 1
Comments and Suggestions for Authors
Line 64: a bit pedantic but I would not consider T. delbrueckii to be the “most closed” species to S. cerevisiae as there are now also a strong focus on Kazachstania spp. in some wine labs. Please reword.
Answer: we modified the sentence
Line 80: is 80% really the final concentration. It is unusually high? Please clarify.
Answer: we made a mistake. We used 40% (corrected in the text)
Line 85: “then” not “the”
Answer: we have corrected in the text
Line 111: Please add more details of the manufacturer of the Baumé (◦Bé) densimeter
Answer: We added some information
Line 114: shouldn’t it be non-S. Cerevisiae strains? Normally S. cerevisiae cannot grow on this medium
Answer: we corrected in the text
Line 120: how many colonies were screened with PCR?
Answer: 10 isolates of S. cerevisiae per fermentation trial (reported in the text)
Line 179: duplication of “in the”
Answer: corrected in the text
Table 1: the 5.685±0.02c, 0.31±0.014a, 5.355±0.03c, 0.225±0.007a and 0.305±0.007a values have three decimal points and the 5.27±0,01c value has a comma
Answer: we corrected in the table
Table 2: in “brackets”, hexanoate, isoamyl alcohol, terpenes (similar typos also in Table 4)
Answer: we correct in the table
Line 221: terpenes throughout the manuscript is wrongly spelled
Answer: we corrected in the text
Section 3.2. Species names not in cursive
Answer: we correct in the text
Table 3: Values 0.25±0.028. 0.23±0.007 and 0.25±0.063 have three decimal points
Answer: we corrected in the text
Figure 3: Please rearrange the order of the descriptors within the spider chart (sensory analysis) as is is a bit difficult to interpret, for instance between “phenolic and sweet toasted it is difficult to know which is which.
Answer: we modify the figure 3 by adding the indicators. We hope that could be sufficient to clarify the results
Line 284: please rephrase the line, it sounds clumsy
Answer: we rewrite the sentence
Line 291: “affected”
Answer: Corrected in the text
some references are missing DOIs (like 9 and 12, 23), 15, 27 have some word starting with a capital letter
Answer: we added the DOIs that are missing and revised the whole references
Hoping our corrections will be satisfactory.
Reviewer 2 Report
Introduction:
Overall the introduction fits the topic of the article well. However, some sentences are not clear and difficult to understand:
1.) Line 57: "In particular, the presence of T. delbrueckii strain in must led to a decrease in wild yeasts biodiversity if compared to the addition of sulphites [19].
What is meant with "in must led"?
2.) Lines 62 and 63: "Effectively, with respect to the attributes required to perform industrial alcoholic fermentation, among non-Saccharomyces 63 yeasts, T. delbrueckii is the most closed species to S. cerevisiae."
What is meant with "closed"? Closely related perhaps?
3.) Line 61: "and others that concern microbial interactions such as the production of active compounds (killer toxin and hydroxytyrosol)."
This is confusing, what do you mean with "others"? Other yeast species? Or other features of T. delbrueckii?
Methods section:
4.) Line 85: "The grapes were the processed using the following procedures: soft hydraulic pressing and cold clarification at 8 ◦C for 2 86 days)."
"then" instead of "the"
5.) Line 88: please define the abbreviation for "YAN"
6.) 2.5. Statistical Analysis
According to the numbering of sections this should be section 2.6
Results:
7.) In figures 1 and 2 only very few data points are shown for the fermentations. This makes the experiments difficult to interpret properly. Please see the article by Puertas et al. (2017) Journal of Applied Microbiology, Volume 122, Issue 3 (0.1111/jam.13375) for an example on how to properly report growth curves of yeasts in the field of winemaking.
8.) Line 171: "In Figure 1 are reported the results of the yeasts viable population during the inoculated fermentations with (on the left) and without (on the right) the addition of 30 mg/L 172 of SO2.
What does left and right mean, figure 1a vs figure 1b? If so, this must be referenced accordingly.
9.) Line 173: "The S. cerevisiae commercial strain OKAY® (Fig. 1a)..."
This does not seem to be correct, according to the legend strain OKAY is not represented in any of the figure panels (the legend says it is strain DiSVA 709).
10.) Figure 2. Indicate that ALPHA is a T. delbrueckii strain.
11.) Figure 2: The growth curves of S. cerevisiae DiSVA 709 look strikingly similar in panels A to C. There should be some natural variation, it almost looks like the same data were used in all panels.
12.) Figure 2: The growth curves of T. delbrueckii look strikingly similar in panels B and C. There should be some natural variation, it almost looks like the same data were used in the two panels.
13.) Table 4. Volatiles: what is the meaning of these data? How do these compounds contribute to the taste of the wine? Readers not familiar in the field would probably not know how to correctly interpret these data. More information would be appreciated.
Overall, at present the data presented by the authors do not convince me, especially given the data shown in figures 1 and 2. Too few data points and too much similarity among several curves appears problematic to me.
Moderate editing of English language is recommended.
Author Response
Reviewer 2
Introduction:
Overall the introduction fits the topic of the article well. However, some sentences are not clear and difficult to understand:
1.) Line 57: "In particular, the presence of T. delbrueckii strain in must led to a decrease in wild yeasts biodiversity if compared to the addition of sulphites [19].
What is meant with "in must led"?
Answer: we modified the sentence
2.) Lines 62 and 63: "Effectively, with respect to the attributes required to perform industrial alcoholic fermentation, among non-Saccharomyces 63 yeasts, T. delbrueckii is the most closed species to S. cerevisiae."
What is meant with "closed"? Closely related perhaps?
Answer: Yes, we made a mistake
3.) Line 61: "and others that concern microbial interactions such as the production of active compounds (killer toxin and hydroxytyrosol)."
This is confusing, what do you mean with "others"? Other yeast species? Or other features of T. Delbrueckii?
Answer: in this sentence, we want to underline that T. delbrueckii is not only important in the wine quality but also for the other characteristics such us its interaction in the production of killer toxins
Methods section:
4.) Line 85: "The grapes were the processed using the following procedures: soft hydraulic pressing and cold clarification at 8 ◦C for 2 86 days)."
"then" instead of "the"
Answer: we corrected in the text
5.) Line 88: please define the abbreviation for "YAN"
Answer: we added the definition of YAN: yeast assimilable nitrogen
6.) 2.5. Statistical Analysis
According to the numbering of sections this should be section 2.6
Answer: we corrected in the text
Results:
7.) In figures 1 and 2 only very few data points are shown for the fermentations. This makes the experiments difficult to interpret properly. Please see the article by Puertas et al. (2017) Journal of Applied Microbiology, Volume 122, Issue 3 (0.1111/jam.13375) for an example on how to properly report growth curves of yeasts in the field of winemaking.
Answer: we added new figures . In particular, we added in figure 2 a new graph we highlighted the statistical variations of wild yeasts in the first two days ( before the sequential inoculum of S. cerevisiae starter strain.
8.) Line 171: "In Figure 1 are reported the results of the yeasts viable population during the inoculated fermentations with (on the left) and without (on the right) the addition of 30 mg/L 172 of SO2.
What does left and right mean, figure 1a vs figure 1b? If so, this must be referenced accordingly.
Answer: we added in the text the letters a and b
9.) Line 173: "The S. cerevisiae commercial strain OKAY® (Fig. 1a)..."
This does not seem to be correct, according to the legend strain OKAY is not represented in any of the figure panels (the legend says it is strain DiSVA 709).
Answer: we corrected the figure legend
10.) Figure 2. Indicate that ALPHA is a T. Delbrueckii strain.
Answer: we have specified in the text
11.) Figure 2: The growth curves of S. cerevisiae DiSVA 709 look strikingly similar in panels A to C. There should be some natural variation, it almost looks like the same data were used in all panels.
Answer: In the same environmental conditions and same substrate the behavior of S. cerevisiae strains is quite similar. The data used come from different biological trials. The cited figures are similar, but not identical. There are some little variations in the data.
12.) Figure 2: The growth curves of T. delbrueckii look strikingly similar in panels B and C. There should be some natural variation, it almost looks like the same data were used in the two panels.
Answer: the same consideration that we reported at point 11. Moreover, the data refer of different strains of T. delbrueckii
13.) Table 4. Volatiles: what is the meaning of these data? How do these compounds contribute to the taste of the wine? Readers not familiar in the field would probably not know how to correctly interpret these data. More information would be appreciated.
Answer: We already enclosed the Active thresholds values in tables 2 and 4. In table 4 we added the Odor active value (OAV) that more easy showed the effect of volatile compound on the analytical and in sensory evaluation .
Overall, at present the data presented by the authors do not convince me, especially given the data shown in figures 1 and 2. Too few data points and too much similarity among several curves appears problematic to me.
Answer: In this regard, we added the analysis of variance of wild yeast population in Fig.2 in the first two days after the inoculation of S. cerevisiae and the two T. delbrueckii strains to highlight the effect of strain inoculation on wild yeast without SO2 addition
Hoping our corrections will be satisfactory.
Reviewer 3 Report
Dear Authors,
I have carefully read the submitted article. The choice of the subject - Biocontrol using Torulaspora delbrueckii in sequential fermen-2 tation: new insights into low sulfites Verdicchio wines – seems to be justified, as the use of non-Saccharomyces yeasts for biocontrol purposes in winemaking is getting more and more interest in recent times.
Overall, the paper is well written. Nevertheless, some crucial issues arise, and some critical information is missing and should be added to make the article suitable. Consequently, I recommend major revisions.
The main drawback of this work is that the chemical parameters of the wines (Table 1 and Table 3) show an unstable situation. First, residual sugars are not reported, and this data should be added in order to make the reader able to understand if differences in ethanol content (first column) are due to incomplete fermentations. Second, and more important, the malic acid content suggests that, most likely, a malolactic fermentation was undergoing in these wines when they were analyzed, and that it was not completed. Moreover, in the first fermentation set, the extent of malic acid consumption was significantly different between trials. Therefore, no speculation can be done by the authors about the contribution of different yeasts to aromatc profiles of wines, as it is widely accepted that the impact of malolactic bacteria on wine aroma is very significant. Indeed, when discussing about differences in aromatic compounds between wines, the Authors cannot rule out the hypothesis that they are due to different MLF progress. New analyses of wines at the time of wine tastings (3 months later) should also be addes, in order to verify if sensory analyses reflect the same situation (at 4°C, if no filtration and no stabilization is done, as in this case, MLF can possibly proceed although slowly).
Another weakness of the study, considering the emphasis of the title, is that the biocontrol action of T. delbrueckii is not flagrant. In the first trial, a quite good result was reached towards wild yeasts in unsulphited conditions (fig.1, panel 3b), but a really similar outcome was achieved through the inoculation of the commercial S. cerevisiae strain alone (depicted in fig.1, panel 1b). In the second trial, one log of difference was detected at day 2 (wild yeasts, difference between fig.2b and 2 a/c); nevertheless, since no other cell counts were made until day 9, the impact of such a difference is hardly assessable. This is also due to the fact that standard deviations (if any) are not shown, and that the number of biological replicates and technical (i.e. plate count) repetitions are never assessed, neither in the methods section nor in the results.
As a more specific comment, Authors should completely rewrite the description of figure 1, making it coherent between figure caption and lines 170-185. For instance, the figure caption does not mention the commercial strain Okay, whereas it is reported in lines 170-185. At the same time, the text describes panels as “fermentations with (on the left) and without (on the right) the addition of 30 mg/L of SO2.” In the subsequent line, although, the difference between left and right of panel 1 seems to be the strain…
Author Response
Reviewer 3
Dear Authors,
I have carefully read the submitted article. The choice of the subject - Biocontrol using Torulaspora delbrueckii in sequential fermen-2 tation: new insights into low sulfites Verdicchio wines – seems to be justified, as the use of non-Saccharomyces yeasts for biocontrol purposes in winemaking is getting more and more interest in recent times.
Overall, the paper is well written. Nevertheless, some crucial issues arise, and some critical information is missing and should be added to make the article suitable. Consequently, I recommend major revisions.
The main drawback of this work is that the chemical parameters of the wines (Table 1 and Table 3) show an unstable situation. First, residual sugars are not reported, and this data should be added in order to make the reader able to understand if differences in ethanol content (first column) are due to incomplete fermentations.
Answer: The fermentations are all finished. We reported in supplemental materials the sugar consumption Fig 1s first trials and Fig 2s second set of fermentation trials and cited in the results.
Second, and more important, the malic acid content suggests that, most likely, a malolactic fermentation was undergoing in these wines when they were analyzed, and that it was not completed. Moreover, in the first fermentation set, the extent of malic acid consumption was significantly different between trials. Therefore, no speculation can be done by the authors about the contribution of different yeasts to aromatc profiles of wines, as it is widely accepted that the impact of malolactic bacteria on wine aroma is very significant. Indeed, when discussing about differences in aromatic compounds between wines, the Authors cannot rule out the hypothesis that they are due to different MLF progress. New analyses of wines at the time of wine tastings (3 months later) should also be addes, in order to verify if sensory analyses reflect the same situation (at 4°C, if no filtration and no stabilization is done, as in this case, MLF can possibly proceed although slowly).
Answer: we are sorry for the mistake. we made a mistake in reporting the data of malic acid in the table. In yellow the modifications. In particular, the data that referred to 0.45 g/L residue was 1.45 g/L indicating that MLF was not occurred. The wines at the end of fermentation were stabilized (filtration and So2 added) but not further analysis were carried out .
Another weakness of the study, considering the emphasis of the title, is that the biocontrol action of T. delbrueckii is not flagrant. In the first trial, a quite good result was reached towards wild yeasts in unsulphited conditions (fig.1, panel 3b), but a really similar outcome was achieved through the inoculation of the commercial S. cerevisiae strain alone (depicted in fig.1, panel 1b). In the second trial, one log of difference was detected at day 2 (wild yeasts, difference between fig.2b and 2 a/c); nevertheless, since no other cell counts were made until day 9, the impact of such a difference is hardly assessable. This is also due to the fact that standard deviations (if any) are not shown, and that the number of biological replicates and technical (i.e. plate count) repetitions are never assessed, neither in the methods section nor in the results.
Answer: In the revision version we added several new information and new elaborations of the data (see Figure 2d) to highlight the effect of the presence of T. delbrueckii. In particular, in sequential fermentation where the control of wild yeast without SO2 is more difficult. The two important aspects that have been pursued are the control of wild yeasts in mixed fermentations without the presence of SO2 and the effects on the sensory analytical profile of the wines
As a more specific comment, Authors should completely rewrite the description of figure 1, making it coherent between figure caption and lines 170-185. For instance, the figure caption does not mention the commercial strain Okay, whereas it is reported in lines 170-185. At the same time, the text describes panels as “fermentations with (on the left) and without (on the right) the addition of 30 mg/L of SO2.” In the subsequent line, although, the difference between left and right of panel 1 seems to be the strain…
Answer: Le figure e le captions of the figures were revised and the mistakes have been corrected
Hoping our corrections will be satisfactory
Reviewer 4 Report
The presented article „Biocontrol using Torulaspora delbrueckii in sequential fermentation: new insights into low sulfites Verdicchio wines” refers to an important goal of the wine industry – to produce high quality wines with health benefits. The research group investigated the application of Torulaspora delbrueckii in the fermentation of wine, especially in combination with a commercial S. cerevisiae species, resulting in distinguished and enhanced flavor and aroma.
The manuscript’s topic is relevant in the food science community. The research shows important benefits from the use of a specific non-Saccharomyces yeast as a biocontrol tool in winemaking business.
In the list of references, only 37% of the publications are from the last 5 years. In my opinion, it would be good if there was a little more discussion on the health benefits of such wines. The authors pointed a lot of self-references (10 out of 37 total).
The manuscript needs a lot of corrections, described below
- According to Author guidelines in Foods, the abstract should be no more than 200 words! Please, try to shorten the abstract.
- Delete the unnecessary intervals between words in rows 43, 47, 77, 79
- Rows 229 – 249: please, write the Latin names in Italic.
- Please, change the numeration of all figures! – You have Figure 2 in Row 180 and Row 232 at the same time!
When you describe Figure 1 in the text, you can make it simple like this: Fig. 1 (a, b, c, d, e, f).
- If you use small letters in Figure 1 (a, b, c, and so on), you should use small letters in Figure 2 as well (not A, B, C).
- Under each figure, please, put Time (days) in a centered position.
- In the references list, the journal titles should be abbreviated according to the guidelines.
- All other recommendations are pointed out in green color in the PDF file.
As a reviewer, I declare that I do not have any kind of conflicts of interest with the authors of this article.

- It would be good if the authors thoroughly check the English language.
Author Response
Reviewer 4
The presented article „Biocontrol using Torulaspora delbrueckii in sequential fermentation: new insights into low sulfites Verdicchio wines” refers to an important goal of the wine industry – to produce high quality wines with health benefits. The research group investigated the application of Torulaspora delbrueckii in the fermentation of wine, especially in combination with a commercial S. cerevisiae species, resulting in distinguished and enhanced flavor and aroma.
The manuscript’s topic is relevant in the food science community. The research shows important benefits from the use of a specific non-Saccharomyces yeast as a biocontrol tool in winemaking business.
In the list of references, only 37% of the publications are from the last 5 years. In my opinion, it would be good if there was a little more discussion on the health benefits of such wines. The authors pointed a lot of self-references (10 out of 37 total).
The manuscript needs a lot of corrections, described below
- According to Author guidelines in Foods, the abstract should be no more than 200 words! Please, try to shorten the abstract.
Answer:
The abstract was revised and reduced . Reported in the text
- Delete the unnecessary intervals between words in rows 43, 47, 77, 79
Answer: we have removed from the text
- Rows 229 – 249: please, write the Latin names in Italic.
Answer: we have corrected in the text
- Please, change the numeration of all figures! – You have Figure 2 in Row 180 and Row 232 at the same time!
When you describe Figure 1 in the text, you can make it simple like this: Fig. 1 (a, b, c, d, e, f).
- If you use small letters in Figure 1 (a, b, c, and so on), you should use small letters in Figure 2 as well (not A, B, C).
- Under each figure, please, put Time (days) in a centered position.
Answer: we modified the figure and corrected in the text
- In the references list, the journal titles should be abbreviated according to the guidelines.
Answer: corrected in the text
- All other recommendations are pointed out in green color in the PDF file.
As a reviewer, I declare that I do not have any kind of conflicts of interest with the authors of this article.
Answer: we revised the text takin g in account the suggestions of the reviewer ( see yellow highlight)
Hoping our corrections will be satisfactory
Reviewer 5 Report
The study is about the utilization of a strain of T. delbrueckii in fermentation with S. cerevisiae aiming for a potential biocontrol of wild yeasts and improvement of aromatic characteristics of the wine. The study presents interesting results about the aromatic alterations induced by the utilization of T. delbrueckii, however, it only explores superficially the biocontrol effect of T. delbrueckii on wild yeasts. There is another study really similar to this one (Agarbati et al, 2020. Applied Sciences, 10:19, 6722), only altering some yeast strains, which reduces the novelty aspect of the present study. I added some corrections and considerations presented below.
Line 28 - The keywords should not be in the title of the manuscript. Please alter the keywords that are already in the title.
Lines 61-62 - Is there some reference that supports this statement about the active compounds produced by T. delbrueckii?
Lines 67-71 - Revise this paragraph, it is confusing and there is information that is repeated.
Line 73 - In this subtitle section you could add the description of two T. delbrueckii utilized.
Line 88 - Add (yeast assimilable nitrogen) to YAN.
Line 90 - Alter “yeast assimilable nitrogen” to “YAN”.
Line 98 - Add the origin of ALPHA the first time that this yeast was cited in the text.
Line 104 - Alter (vol/vol) to (v/v).
Lines 120-126 - A brief description of how the sequencing was made could be added. Also, this data was not explored in the results section. How many different species were visualized? Did you count each conjunct of colonies with particular morphology or only enumerate colonies that were not S. cerevisiae or T. delbrueckii as a unique group?
Lines 171-185 - When I first see Fig1 I could not understand what each graphic was, so I expected to read the text and understand the Fig1. Well, I still am not sure what graphic is about OKAY, DiSVA 709, or DiSVA 130/DiSVA 709. In the text, some graphics are referenced, while others, like in line 174, is calling only a letter (Fig. b) and some are not even mentioned. So, revise this paragraph, also what happened in Fig 3a? Why SO2 did not reduce the population of wild yeasts in Fig3 like in Fig 1a and Fig 2a?
When the plates were evaluated how many species of wild yeast were observed, and were they identified?
In the Material and Methods section, the authors mention that the fermentations were monitored by the evaluation of sugar consumption, so I expected this data. They could be added to the manuscript and will help to visualize how much sugar was consumed by the T. delbrueckii at the start of the fermentation, or when the fermentation ended.
Line 176 - Be careful when you use the term “disappearing” because what normally happen is that these yeasts are not detected by the dilutions that are plated and counted. So is better to say that these yeasts are not detected.
Line 179 - There are two “in the”.
Lines 182-185 - Be more clear is this sentence. Are the percentages related to the proportion of inoculated S. cerevisiae in comparison with wild S. cerevisiae?
Fig 1 - This figure is really hard to understand, the inner numbers (1a; 1b...) don’t are described in the legend, so the reader could only guess what they represent. Also, the utilization of the format “1a...” cause confusion along the text, because when you reference “Fig 2a” I expect this to be in the second figure and not be a reference to the first one. I suggest a change in how each graphic is called, utilizing only letters, and please, add in the legend what each graphic represents. The graphic “2a” has different formatting.
Table 1 - Add the information about the statistical analysis utilized in the table, what the letters represent, the significance level, and how I should compare the data (columns or lines).
Line 207 - What are this thresholds values presented?
Table 2 - Same problem that is in Table 1. Alter “Ethyl exanoate” to “Ethyl hexanoate”.
Lines 227-228 - I revise Fig1, but I cannot see how this strain of T. delbrueckii controls wild yeasts. What I see is that the wild yeast population does not increase in graphic “3b”, but why this population increases in graphic “3a”?
Line 234 - How much was that reduction?
Figure 2 - There are some parts of the graphics that are in gray, and Fig 2C has different formatting.
Table 3 - Same problem with the statistical information. The concentration of geraniol stated as “0.00” could be represented as “ND”, similar to Table 2.
Line 268 - The utilization of the “second fermentation” expression is problematic for me because it reminds me of the sparkling wine production process.
Line 273 - Why in the text is written “ripe tropical” and in Fig3 “Tropical fruit”?
In the Discussion section, the authors explored poorly the mechanisms of how T. delbrueckii could control another yeast. Please add more information about this.
Revise the text. There are some random spaces along the text that could be removed (e.g., lines 42 and 47). Also, some scientific names are not in italic format. Please ensure consistency with the utilization of “sulfite” or “sulphite”.
The study could be improved, the authors emphasized “biocontrol using T. delbrueckii” in the title, but this aspect was poorly explored throughout the manuscript. I believe that the first assay lacks two essential controls that would strengthen the main hypothesis of the study. Firstly, the authors should have included a control without any inoculation. This control would demonstrate how much the wild yeast population could grow without any intervention. Secondly, a fermentation using only the T. delbrueckii strain is missing. This control would show how much this yeast could control the wild yeasts during fermentation. I understand that these two fermentation would produce a really bad wine or not even conclude the fermentation process. However, they would help to understand the real potential of T. delbruekii in controlling other yeasts.
Author Response
Reviewer 5
The study is about the utilization of a strain of T. delbrueckii in fermentation with S. cerevisiae aiming for a potential biocontrol of wild yeasts and improvement of aromatic characteristics of the wine. The study presents interesting results about the aromatic alterations induced by the utilization of T. delbrueckii, however, it only explores superficially the biocontrol effect of T. delbrueckii on wild yeasts. There is another study really similar to this one (Agarbati et al, 2020. Applied Sciences, 10:19, 6722), only altering some yeast strains, which reduces the novelty aspect of the present study. I added some corrections and considerations presented below.
Line 28 - The keywords should not be in the title of the manuscript. Please alter the keywords that are already in the title.
Answer: two key words were modified but we left T. DELBRUECKII
Lines 61-62 - Is there some reference that supports this statement about the active compounds produced by T. Delbrueckii?
Answer: we added two new references but in the discussion section as one of the possible explanation of the biocontrol action
(Villalba, M.L.; Sáez, J.S.; Del Monaco, S.; Lopes, C.A.; Sangorrín, M.P. TdKT, a new killer toxin produced by Torulaspora del‐ brueckii effective against wine spoilage yeasts. Int. J. Food Microbiol. 2016, 217, 94–100. Ramírez, M.; Velázquez, R.; Maqueda, M.; Martínez, A. Genome organization of a new double‐stranded RNA LA helper virus from wine Torulaspora delbrueckii killer yeast as compared with its Saccharomyces counterparts. Front. Microbiol. 2020, 11, 2977)
Lines 67-71 - Revise this paragraph, it is confusing and there is information that is repeated.
Answer: we modified the sentence
Line 73 - In this subtitle section you could add the description of two T. Delbrueckii utilized.
Answer: we reported the citations of published papers for more information on DiSVA 130 while the other is a commercial strain
Line 88 - Add (yeast assimilable nitrogen) to YAN.
Answer: we corrected in the text
Line 90 - Alter “yeast assimilable nitrogen” to “YAN”.
Answer: Answer: we corrected in the text
Line 98 - Add the origin of ALPHA the first time that this yeast was cited in the text.
Answer: in the text we reported the commercial information of this starter strain and its use in wine making for the production of wines with a marked aromatic complexity characterized by persistence and fullness
Line 104 - Alter (vol/vol) to (v/v).
Answer: we modified in the text
Lines 120-126 - A brief description of how the sequencing was made could be added. Also, this data was not explored in the results section. How many different species were visualized? Did you count each conjunct of colonies with particular morphology or only enumerate colonies that were not S. cerevisiae or T. delbrueckii as a unique group?
Answer: The detection and enumeration of inoculated and wild yeasts were evaluated to combine the results of lysine agar, and the analysis of macro- and micro-morphological colonies in WL nutrient agar medium. The colonies that were not S. cerevisiae or T. delbrueckii were enumerated as wild yeasts but micro-macro morphological evaluation indicated that from 60 to 80% were belonging to Hanseniaspora uvarum species.
Lines 171-185 - When I first see Fig1 I could not understand what each graphic was, so I expected to read the text and understand the Fig1. Well, I still am not sure what graphic is about OKAY, DiSVA 709, or DiSVA 130/DiSVA 709. In the text, some graphics are referenced, while others, like in line 174, is calling only a letter (Fig. b) and some are not even mentioned. So, revise this paragraph, also what happened in Fig 3a?
Answer: we modified the figure and corrected in the text
Why SO2 did not reduce the population of wild yeasts in Fig3 like in Fig 1a and Fig 2a?
When the plates were evaluated how many species of wild yeast were observed, and were they identified?
Answer Actually after a reduction of wild yeasts due to the dose of SO2 a more resistant species could develop. Typically between 60 and 80% were easily detectable colonies on WL agar. At time 0, however, a species resistant to the SO2 dose may have developed but was then controlled by the inoculated cultures. In the specific time we did not differentiate the enumerated wild yeast species
In the Material and Methods section, the authors mention that the fermentations were monitored by the evaluation of sugar consumption, so I expected this data. They could be added to the manuscript and will help to visualize how much sugar was consumed by the T. delbrueckii at the start of the fermentation, or when the fermentation ended.
Answer: The sugar consumption of the two set of fermentation trials was reported in Figure 1s and figure 2s in supplemental materials
Line 176 - Be careful when you use the term “disappearing” because what normally happen is that these yeasts are not detected by the dilutions that are plated and counted. So is better to say that these yeasts are not detected.
Answer: we corrected in the text
Line 179 - There are two “in the”.
Answer: we corrected in the text
Lines 182-185 - Be more clear is this sentence. Are the percentages related to the proportion of inoculated S. cerevisiae in comparison with wild S. Cerevisiae?
Answer: we added the range of the S. starter yeasts 60-90%
Fig 1 - This figure is really hard to understand, the inner numbers (1a; 1b...) don’t are described in the legend, so the reader could only guess what they represent. Also, the utilization of the format “1a...” cause confusion along the text, because when you reference “Fig 2a” I expect this to be in the second figure and not be a reference to the first one. I suggest a change in how each graphic is called, utilizing only letters, and please, add in the legend what each graphic represents. The graphic “2a” has different formatting.
Answer: we change and corrected the figure and the manuscript accordingly
Table 1 - Add the information about the statistical analysis utilized in the table, what the letters represent, the significance level, and how I should compare the data (columns or lines).
Answer: we corrected in the text and Tables
Line 207 - What are this thresholds values presented?
Answer: the threshold values represent the minimum concentrations that can be perceived at the gustatory level. We added in Table 3 odor active values (OAV) also
Table 2 - Same problem that is in Table 1. Alter “Ethyl exanoate” to “Ethyl hexanoate”.
Answer: corrected in the text
Lines 227-228 - I revise Fig1, but I cannot see how this strain of T. delbrueckii controls wild yeasts. What I see is that the wild yeast population does not increase in graphic “3b”, but why this population increases in graphic “3a”?
Answer: At time 0, as reported above, a species resistant to the SO2 dose may have developed but it was then controlled by the inoculated cultures. In the specific time we did not differentiate the enumerated wild yeast species but it is possible that selected species by SO2 could be more resistant to the T. delbrueckii strain. This specific situation could be explain this behavior.
Line 234 - How much was that reduction?
Answer: the reduction is about 0.8% v/v
Figure 2 - There are some parts of the graphics that are in gray, and Fig 2C has different formatting.
Answer: we corrected in the text
Table 3 - Same problem with the statistical information. The concentration of geraniol stated as “0.00” could be represented as “ND”, similar to Table 2.
Answer: we corrected in the text
Line 268 - The utilization of the “second fermentation” expression is problematic for me because it reminds me of the sparkling wine production process.
Aswer: we added “second set of fermentation”
Line 273 - Why in the text is written “ripe tropical” and in Fig3 “Tropical fruit”?
Aswer: we corrected in the text
In the Discussion section, the authors explored poorly the mechanisms of how T. delbrueckii could control another yeast. Please add more information about this.
Answer: Some hypothesis of biocontrol action of T. delbrueckii strain in the discussion section are enclosed adding some references
Revise the text. There are some random spaces along the text that could be removed (e.g., lines 42 and 47). Also, some scientific names are not in italic format. Please ensure consistency with the utilization of “sulfite” or “sulphite”.
Answer: corrected in the text
The study could be improved, the authors emphasized “biocontrol using T. delbrueckii” in the title, but this aspect was poorly explored throughout the manuscript. I believe that the first assay lacks two essential controls that would strengthen the main hypothesis of the study. Firstly, the authors should have included a control without any inoculation. This control would demonstrate how much the wild yeast population could grow without any intervention. Secondly, a fermentation using only the T. delbrueckii strain is missing. This control would show how much this yeast could control the wild yeasts during fermentation. I understand that these two fermentation would produce a really bad wine or not even conclude the fermentation process. However, they would help to understand the real potential of T. delbruekii in controlling other yeasts
Answer: Th aim of this work is finalized towards the use of mixed fermentation with improved of some characters of the wines ( aroma complexity, aromatic and sensorial profiles …) in a context of SO2 reduction. In this regard, the effect of no addition of SO2 at the inoculation time could of interest specifically in mixed fermentation. In the revised version we added the figure 2D where the analysis of variance of time 0 and time 2 days of wild yeasts was done.
Some hypothesis of biocontrol action of T. delbrueckii strain in the discussion section are enclosed
Hoping our corrections will be satisfactory.
Round 2
Reviewer 2 Report
The authors have addressed my previous comments adequately. The data that is presented is more convincing now. Overall, the improvements are significant so that the manuscript is now publishable in Foods.
Although I am not a native speaker of English I´d say overall the use of English is fine. Still, moderate English editing would improve the quality of the manuscript.